# Development and Optimization of a Silica Column-Based Extraction Protocol for Ancient DNA

**DOI:** 10.3390/genes13040687

**Published:** 2022-04-13

**Authors:** Marianne Dehasque, Patrícia Pečnerová, Vendela Kempe Lagerholm, Erik Ersmark, Gleb K. Danilov, Peter Mortensen, Sergey Vartanyan, Love Dalén

**Affiliations:** 1Centre for Palaeogenetics, Svante Arrhenius väg 20C, 10691 Stockholm, Sweden; vendela.kempe.lagerholm@arklab.su.se (V.K.L.); erik.ersmark@nrm.se (E.E.); love.dalen@nrm.se (L.D.); 2Department of Bioinformatics and Genetics, Swedish Museum of Natural History, P.O. Box 50007, 10405 Stockholm, Sweden; 3Department of Zoology, Stockholm University, 10691 Stockholm, Sweden; 4Section for Computational and RNA Biology, Department of Biology, University of Copenhagen, 2200 Copenhagen, Denmark; patricia.pecnerova@bio.ku.dk; 5Department of Archaeology and Classical Studies, Stockholm University, Lilla Frescativägen 7, 11418 Stockholm, Sweden; 6Peter the Great Museum of Anthropology and Ethnography, Kunstkamera, Russian Academy of Sciences, University Embankment 3, Saint-Petersburg P.O. Box 199034, Russia; gleb.danilov.spb@gmail.com; 7Department of Zoology, Swedish Museum of Natural History, P.O. Box 50007, 10405 Stockholm, Sweden; peter.mortensen@nrm.se; 8North-East Interdisciplinary Scientific Research Institute N.A.N.A. Shilo, Far East Branch, Russian Academy of Sciences, Magadan 68500, Russia; sergey-vartanyan@mail.ru

**Keywords:** ancient DNA, woolly mammoth, DNA extraction, high-throughput sequencing, bone

## Abstract

Rapid and cost-effective retrieval of endogenous DNA from ancient specimens remains a limiting factor in palaeogenomic research. Many methods have been developed to increase ancient DNA yield, but modifications to existing protocols are often based on personal experience rather than systematic testing. Here, we present a new silica column-based extraction protocol, where optimizations were tested in controlled experiments. Using relatively well-preserved permafrost samples, we tested the efficiency of pretreatment of bone and tooth powder with a bleach wash and a predigestion step. We also tested the recovery efficiency of MinElute and QIAquick columns, as well as Vivaspin columns with two molecular weight cut-off values. Finally, we tested the effect of uracil-treatment with two different USER enzyme concentrations. We find that neither bleach wash combined with a predigestion step, nor predigestion by itself, significantly increased sequencing efficiency. Initial results, however, suggest that MinElute columns are more efficient for ancient DNA extractions than QIAquick columns, whereas different molecular weight cut-off values in centrifugal concentrator columns did not have an effect. Uracil treatments are effective at removing DNA damage even at concentrations of 0.15 U/µL (as compared to 0.3 U/µL) of ancient DNA extracts.

## 1. Introduction

Ancient DNA is a powerful tool to study the past. Much progress has been made in recent years in ancient DNA research, mainly due to the introduction of high-throughput sequencing (HTS) technologies, as well as improvements in laboratory and bioinformatic methods [1,2,3]. It is now possible to generate whole-genome data from extinct lineages such as woolly mammoths (*Mammuthus primigenius*) and Neanderthals (*Homo neanderthalensis*) [4,5], study population dynamics based on whole-genome data of samples several thousand years old [6,7], and generate draft mitochondrial and nuclear genomes of samples dating to the Middle and Early Pleistocene [8,9,10].

Despite these accomplishments, there are still several challenges when working with ancient DNA. Due to degradation processes over time, ancient DNA is usually preserved in low amounts in the sample [11,12,13]. Contamination from modern sources, such as soil bacteria, people handling the samples, or contaminated reagents, can easily overwhelm the endogenous ancient DNA [4,14]. Furthermore, ancient DNA is typically degraded into DNA fragments of less than 100 base pairs [13,15] and is characterized by post-mortem damage patterns. The most common chemical modification is the deamination of cytosine to uracil, which mainly occurs at fragment ends and results in C- > T or G- > A conversions during sequencing library preparation [16,17].

A cornucopia of extraction methods has been developed to tackle these problems. These methods typically focus on retrieving ultra-short DNA fragments [6,8,18] and reducing external contamination [19,20,21,22,23]. Recently, methods using chemical pretreatment of bone powder to increase the fraction of endogenous DNA have gained increased attention. The reasoning behind these methods is that endogenous DNA is more tightly bound to the bone matrix than exogenous contaminant DNA and is thus less susceptible to pretreatment methods [24,25]. However, the growing number of methods indicates that there is no one-size-fits-all method, and that the quality of the DNA, as well as study goals, are important considerations when choosing a method.

Yang and colleagues [26] developed the first silica spin-column based extraction protocol optimized for ancient DNA. The method has been popular due to its relatively short hands-on time and low cost. However, since then, the original protocol has been modified multiple times, and different variations are in circulation. Common modifications include the replacement of the sodium dodecyl sulfate (SDS) in the digestion buffer with 1 M urea [27], the replacement of Centricon with Amicon Ultra [28] or Vivaspin centrifugal filters [29] to concentrate the digestion lysate, and the replacement of QIAquick spin columns with MinElute spin columns [30]. However, many of these modifications are based on personal experience rather than systematic testing.

The replacement of QIAquick columns with MinElute columns, for example, is based on the fact that MinElute columns are advertised as retaining shorter fragments than QIAquick columns. QIAquick columns are designed to retain fragments of 100 base pairs and above, whereas MinElute columns supposedly retain fragments as short as 70 base pairs. As ancient DNA is typically fragmented, this adaptation thus seems warranted. However, to the best of our knowledge, there have not yet been any studies to test whether changing silica column types results in higher ancient DNA yields.

In a similar fashion, some studies have been using centrifugal filters with molecular weight cut-offs (MWCO) of 30 kDA to concentrate the DNA lysate [28,31,32], corresponding to double-stranded DNA fragment lengths of approximately 50 base pairs. This cut-off value was based on the retrieval of PCR fragments. However, shotgun sequencing and current library preparation methods [33,34] allow for retrieval of even shorter DNA fragments. One could therefore hypothesize that centrifugal concentrator filters with lower molecular weight cut-offs will retain more short DNA fragments and would be more suitable for sequencing with ancient DNA.

Another example of modifications where little systematic testing has been conducted concerns the removal of deaminated cytosine sites in DNA. One way to deal with the damage patterns in ancient DNA is the treatment of extracts with USER enzyme (New England Biolabs), which is a mixture of uracil–DNA–glycosylase (UDG) and endonuclease VIII (endoVIII). First, uracil residues are removed by UDG, resulting in abasic sites. Next, these abasic sites are cleaved with endoVIII, thus effectively removing damage from the ancient DNA fragments [35]. Despite the USER enzyme being costly, amounting to up to 50% of the DNA extraction cost, there is little documentation on how much of the enzyme is needed to effectively remove DNA damage in ancient samples. For example, some studies used 0.3 Units USER enzyme per µL of ancient extract [36,37], whereas others only used half that amount [30,38].

The objective of this study was to develop and optimize a silica column-based extraction protocol, building on the original method presented by [26], where improvements were compared in controlled experiments. To do this, we investigated the efficiency of previously described chemical and enzymatic pretreatment methods. We also tested whether there is a difference in DNA yields between QIAquick and MinElute spin columns (both Qiagen), and whether there is a difference after concentrating the DNA lysis using Vivaspin columns with molecular cut-off weights of 30 and 10 kDa (corresponding to 50 and 20 bp size cut-offs). Finally, we tested two different USER concentrations and their effect on DNA damage patterns. With this new protocol, we aimed to maximize cost and time efficiency, as well as ancient DNA yield.

## 2. Materials and Methods

### 2.1. Sample Information

Bones and tusks from 38 woolly mammoth (*Mammuthus primigenius*) specimens were collected in Siberia on Wrangel Island, Chukotka, and the Taimyr Peninsula (Appendix A). While not all samples were dated, the samples that were radiocarbon dated had age ranges between 4.4 and 31.7 thousand calibrated years before present.

### 2.2. Sampling

We carried out all laboratory procedures in the designated ancient DNA lab facilities at the Swedish Museum of Natural History and at the Centre for Palaeogenetics, both in Stockholm. Lab work was performed using standard ancient DNA lab procedures, including wearing protective suits and facemasks, regular cleaning of all working surfaces with sodium hypochlorite, and UV sterilization of tools and working hoods. Negative extraction controls were added for at least every 7th sample to monitor for contamination introduced during lab handling or present in the reagents, as well as cross-sample contamination.

For all samples, we removed the surface with an electric drill (Dremel, Mt. Prospect, IL, USA) to minimize external contamination. Bone powder was collected by drilling at low speed [39]. For the USER, centrifugal concentrator filters, and silica column experiments, we collected approximately 50 mg of bone powder per treatment. For the predigestion experiment, approximately 100 mg was used, whereas approximately 200 mg of bone/tusk powder was collected for the experiment with sodium hypochlorite (bleach) wash followed by predigestion. To reduce within-sample variability, bone powder was collected during one drilling session and mixed in a single tube by thoroughly shaking it. The bone powder was subsequently subdivided into two equal parts using a precision balance and stored in separate tubes that were used for each comparative analysis.

### 2.3. Extraction Experiment

The standard extraction protocol, which serves as the control experiment in this study, is an in-house method that has been based on the protocols described in [26,29]. Approximately 50 mg of bone/tooth powder is subjected to 715 µL digestion buffer containing 630 µL EDTA (pH 8, 0.5 M), 70 µL Urea (1 M) and 15 µL proteinase K (10 µg/µL) and incubated under motion at 55 °C overnight. Next, samples are centrifuged at 2300 rpm for 5 min, and the supernatant is collected and concentrated using Vivaspin filters with a MWCO of 30 kDa (Sartorius, Göttingen, Germany) by centrifuging at 12,000 rpm until less than 120 µL of the supernatant is left. Samples are subsequently purified using QIAquick spin columns (Qiagen, Hilden, Germany) following the manufacturer’s instructions, with the following exceptions: (1) After each centrifuge step, the QIAquick column is placed in a clean collection tube. (2) After the additional centrifuge step (“Step 7” in the manufacturer’s purification protocol), the QIAquick column is placed in a clean LoBind tube (Eppendorf, Hamburg, Germany) with an open lid for 5 min to remove residual ethanol. (3) DNA is eluted twice with 50 µL of EB buffer to obtain a final volume of ca. 96 µL. Finally, 20 µL of ancient extract is incubated with 6 Units of USER enzyme (New England Biolabs, Ipswich, MA, USA) for 3 h at 37 °C as described in [37].

We tested five different treatments to optimize this in-house silica-based extraction protocol: (1) the addition of a predigestion step; (2) the addition of a washing step with sodium hypochlorite (bleach), followed by a predigestion step; (3) carrying out clean-up steps with different silica columns; (4) using centrifugal concentrator filters with different molecular cut-off values; and (5) testing different concentrations of USER enzyme during uracil treatment. For each of the five treatments, which are described below, samples were divided into two subsamples, and extraction was performed twice, once as described in the original protocol above and once as described in the treatment below.

#### 2.3.1. Predigestion

Prior to the start of the standard extraction protocol, we subjected the samples (*n* = 8) to one additional short digestion (i.e., predigestion) step. We added 715 µL digestion buffer containing 0.5 M EDTA (pH 8), 0.1 M Urea and 15 µL proteinase K (10 µg/µL) to the samples and incubated them under motion for 30 min at 37 °C. The supernatant was removed from the sample after centrifuging at 2300 rpm for 5 min. After this predigestion step, the experiment was continued with the overnight digestion and extraction as described in the standard protocol.

#### 2.3.2. Bleach Wash and Predigestion

We washed the bone/tooth powder (*n* = 7) with 0.5% sodium hypochlorite for 10 min under motion at room temperature. The supernatant was removed after centrifuging the sample at 2300 rpm for 2 min. Next, residual sodium hypochlorite was removed by washing the sample three times with water. For each washing step, 1 mL of UltraPure DNase/RNase-Free distilled water (Invitrogen) was added to the sample. The sample was thoroughly vortexed [22], and water was removed after centrifuging for 2 min at 2300 rpm, and 5 min at 2300 rpm for the final washing step. Next, we carried out one additional predigestion step as described in Section 2.3.1 and extraction with overnight digestion as described in the standard protocol above.

#### 2.3.3. Silica Columns

For 7 bone and tusk powder samples, the standard extraction protocol was performed twice: once with QIAquick spin columns (as in the original protocol) and once with MinElute spin columns (both from Qiagen).

#### 2.3.4. Centrifugal Concentrator Filters

For 10 bone and tusk powder samples, the standard extraction protocol was performed twice, once with Vivaspin filters with a MWCO of 30 kDa (as in the original protocol) and once with Vivaspin filters with a MWCO of 10 kDA (both from Sartorius), corresponding respectively to filtration cut-offs of double-stranded DNA fragment lengths of approximately 50 and 20 base pairs.

#### 2.3.5. USER

For 8 bone and tusk powder samples, the standard extraction protocol was performed once. Next, the extract was split into two parts, and each part was treated with a different concentration of USER enzyme; once with 6 Units of USER enzyme (as in the original protocol) and once with 3 Units of USER enzyme. Following [37], uracil treatment was combined with blunt-end repair as described in “step 4” of the protocol of [33].

### 2.4. Library Preparation and Sequencing

We prepared double-stranded Illumina libraries following the protocol of Meyer and Kircher [33], with the exception that reaction volumes were halved and purification steps were carried out with MinElute columns (Qiagen).

Indexing polymerase chain reactions (PCR) were performed in 25 µL reaction volumes containing 1× AccuPrime reaction mix (Life Technologies, Carlsbad, CA, USA), 0.3 µM of each indexing primer, 1.25 U AccuPrime Pfx DNA polymerase (Life Technologies) and 3 µL of template DNA. The PCR was run with the following reaction conditions: 95 °C for 2 min, followed by initially 12 cycles of 95 °C for 15 s, 60 °C for 30 s and 68 °C for 30 s. We assessed PCR success by running the indexed libraries on an agarose gel. If no product was visible, the amount of indexing cycles was increased to 14 or 16 cycles. Next, we cleaned and size-selected the indexed libraries with Agencourt AMPure XP beads (Beckman Coulter, Brea, CA, USA). The concentrations of the indexed libraries were quantified using a high sensitivity chip on a BioAnalyzer 2100 (Agilent, Santa Clara, CA, USA) and pooled in equimolar ratios. Libraries were sequenced using Illumina NovaSeq technology at the National Genomics Infrastructure (Science for Life Laboratory, Stockholm, Sweden), using paired-end 2 × 100 bp or 2 × 150 bp settings, resulting in an average sequencing effort of 24.3 million reads per sample.

### 2.5. Data Processing and Analysis

Differences in sequencing depth can affect complexity (i.e., the proportion of unique sequencing reads) estimates, with smaller sequencing depths seemingly resulting in higher complexity [40]. To account for these biases and to allow for direct comparison between treatment and control, we downsampled the sequencing files to match the lowest number of sequences for each pair of subsamples using seqtk (https://github.com/lh3/seqtk, accessed on 18 June 2018). The GenErode pipeline [41] written in Snakemake version 4.5.0 [42] was used to subsequently process the sequencing files. In brief, paired-end reads were merged, and adapters were trimmed with default parameters using SeqPrep v1.2 (https://github.com/jstjohn/SeqPrep, accessed on 18 March 2016), and fragments shorter than 30 bp were excluded. A minor modification was made to the source code to calculate the quality score in the merged region [5]. Next, the merged fragments were mapped against a concatenated nuclear–mitochondrial reference genome consisting of the African savannah elephant genome (LoxAfr4, Broad Institute) and the woolly mammoth mitochondrial genome (Genbank accession no. DQ188829 [43]) to avoid mitochondrial-like nuclear segments (NUMTs) mapping to the mitochondrial genome. Mapping was performed using BWA aln v0.7.17 [44], using settings adapted for ancient DNA as described in [37]. Only nuclear fragments with a mapping quality of at least 30 were retained using SAMtools v1.8 [45]. Duplicates were removed using a custom script from [5] that removes duplicates based on both starting and end positions of the read. Indels were realigned using GATK v3.8-0 [46]. Summary statistics were obtained using SAMtools v1.8 [45] and QualiMap v2.2.1 [47].

For each sample, we compared the following parameters between control and treatment: endogenous DNA content (defined as the number of reads mapping to the reference genome divided by the total number of reads before duplicate removal), average fragment length, GC-content of mapped DNA fragments, complexity (here defined as the proportion of uniquely mapping reads after duplicate removal), and genome-wide coverage per kilobase (i.e., per 1000 bp). Coverage depends on all other parameters and thus directly reflects success in a shotgun sequencing experiment. We assessed normality using the Shapiro–Wilk normality test and conducted statistical analyses using either one-sided paired *t* test or Wilcoxon signed test if normality was rejected. To test for any potential damage biases caused by the different treatments, we quantified damage patterns in a Bayesian manner with mapDamage v2.0.9 [48]. More specifically, we determined the probability of cytosine deamination in double-stranded context (δ_D_), in single-stranded context (δ_S_), and the probability of a base terminating in an overhang (λ). Samples with less than 1% endogenous content were removed from analyses, as we cannot rule out that a major proportion of mapped reads in these samples are in spuriously mapping contaminant sequences [49].

## 3. Results

After excluding samples with less than 1% endogenous content and downsampling, between 5.66 and 61.9 million reads were obtained per sample (average 19.7 million). Endogenous content (measured as the fraction of mapping reads divided by the total number of reads) ranged from 1.56–63.0% (average 22.0%). As expected, all samples had an average fragment length smaller than 100 bp [15] (see Appendix A for an overview of the sequencing results and summary statistics). Since the samples were USER treated, overall deamination damage patterns were low (average probability of 0.00621 in double-stranded context and 0.0795 in single-stranded context), although C- > T and G- > A conversions were slightly higher in the two terminal bases (Appendix A). One exception is sample MD123, which was part of the silica column experiment. Despite the extract from the QIAquick and MinElute treatment being USER treated on two different occasions, damage patterns were in both cases high and reminiscent of non-USER treated samples (Appendix A). Further inspection also revealed that average fragment length (average 43.3 bp) and GC content (average 30.8%) were low compared to the other samples (Appendix A).

### 3.1. Predigestion

After data processing, six out of eight samples were retained for the predigestion experiment. Endogenous content (measured as the fraction of mapped reads divided by the total number of reads) was significantly higher after predigestion (t = −3.13 DF = 5, *p* value = 0.0130). However, predigestion also caused a significant loss in complexity (t = 4.06, DF = 5, *p* value = 0.00497) and average fragment length (t =2.02, DF = 5, *p* value = 0.0497, Appendix A). As a result, coverage after predigestion was not significantly higher, although there was a positive trend (t = −1.56, DF = 5, *p* value = 0.0895, Figure 1 and Figure 2). Predigestion did not affect GC content of the mapped reads (V = 9.5, *p* value = 0.624), or any of the measured damage parameters (Table 1, Appendix A).

### 3.2. Bleach and Predigestion

All seven samples were retained for the bleach + predigestion experiment. Although non-significant, six out of seven samples showed a trend toward higher endogenous content after bleach + predigestion (t = −1.75, DF = 6, *p* value = 0.065). However, complexity was significantly lower after bleach + predigestion (t = 5.16, DF = 6, *p* value = 0.00105). In contrast to the predigestion treatment, average fragment length did not differ (t = −0.16, DF = 6, *p* value = 0.438, Appendix A), but GC content of mapped reads was significantly lower (t = 4.51, DF = 6, *p* value = 0.00202). As a consequence of these opposing effects, coverage was not significantly higher after bleach + predigestion (t = −1.33, DF = 6, *p* value = 0.116, Figure 1 and Figure 2). Bleach + predigestion did however significantly increase the probability of cytosine deamination in single-stranded context (δ_S_; t = −8.28, DF = 6, *p* value = 0.0000841), double-stranded context (δ_D_; t = −4.91, DF = 6, *p* value = 0.00134), and the probability of DNA fragments ending in an overhang (λ; t = −8.28, DF = 6, *p* value = 0.00450, Appendix A).

### 3.3. Silica Columns

After data processing, five samples had an endogenous content >1% and were retained. Complexity and endogenous content did not significantly differ between MinElute and QIAquick columns (Table 1, Figure 1 and Figure 2), whereas MinElute columns resulted in significantly higher average fragment length (t = −2.35, DF = 4, *p* value = 0.0392) and significantly lower GC content (V = 15, *p* value = 0.0312). Nevertheless, coverage did not significantly differ between the two silica column treatments (V = 5, *p* value = 0.313). The probability of cytosine deamination did not differ for either single- or double-stranded context (Table 1, Appendix A). However, the probability of DNA fragments ending in an overhang was significantly lower for MinElute columns (λ; V = 15, *p* value = 0.0313, Appendix A).

### 3.4. Centrifugal Concentrator Filters

Seven out of ten samples were retained in the centrifugal concentrator filters experiment. Average fragment length was significantly lower in the filters with 10 kDA cut-off values (i.e., the filters with lower cut-off values) (t = 2.01, DF = 6, *p* value = 0.0461). Furthermore, there was a trend toward the 10 kDA filters retaining more endogenous DNA (t = −1.82, DF = 6, *p* value = 0.0597). However, no difference was found in complexity, GC-content, or coverage between the two filters (Table 1, Figure 1 and Figure 2). The probability of cytosine deamination in single-stranded context in the 10 kDA was significantly higher (δ_S_; V = 12, *p* value = 0.0391, Appendix A), whereas the probability of cytosine deamination in double-stranded context or the probability of ending in a single-stranded overhang showed no difference (Table 1, Appendix A).

### 3.5. USER

Only three samples had an endogenous content >1% and were retained for further analysis. Due to the small sample size, no statistical analysis was feasible, and the results reported here are therefore descriptive. Halving the USER concentration from 6 to 3 Units did not affect endogenous content, GC content, or complexity (Figure 3). Reduced USER concentration did however increase the average fragment length for all three samples, with a relative average increase of 3% (or 1.03-fold) compared to the higher USER concentration (Figure 3, Appendix A). Whereas two samples had higher coverage after half USER treatment, one sample had lower coverage. The samples treated with lower USER concentrations did not differ for damage parameters δ_D_ or λ. However, the probability of cytosine deamination in single-stranded overhangs (δ_S_) was higher for all three samples treated with the lower USER concentration, resulting in a relative increase in probability of 11% (or 1.11-fold) (Appendix A).

## 4. Discussion

We here present an optimized protocol for ancient DNA extraction (Appendix A) based on a series of optimization experiments. In these experiments, we found no clear evidence that chemical or enzymatic pretreatments of bone powder increase the final coverage. This is in contrast with studies that reported increased sequencing efficiency, measured as either the amount of uniquely mapped reads, endogenous content, or library complexity after decontamination pretreatments [19,20,21,22]. Our interpretation of these result is that while bleach wash and/or predigestion may increase the fraction of endogenous content in a sample, it potentially also comes with a trade-off by reducing the total amount of available endogenous DNA [20,23,50,51], with previous studies reporting losses of up to 75% of total DNA [50]. Low amounts of endogenous DNA can in turn affect complexity, i.e., the amount of uniquely mapped reads. Our study shows that although both the predigestion and bleach + predigestion treatments resulted in higher endogenous DNA content, these treatments also caused lower library complexity compared to the control treatments.

It is important to consider sequencing depth when interpreting the results of pretreatment methods. While the reduced library complexity has a limited effect on coverage at low sequencing depths, as demonstrated by the higher coverage of multiple pretreated samples at low sequencing depths (Figure 1), complexity will become more important as compared to endogenous content at high sequencing depths (see Figure 4 for a conceptual example). If the main aim of a study is to maximize coverage at high sequencing depths (>100 million on-target sequencing reads), higher complexity is probably more desirable than the increase in endogenous content. Conversely, some studies and/or samples might benefit from extraction protocols optimized for endogenous DNA content and not complexity, such as studies using low-depth sequencing data (e.g., [52]), or for the removal of contaminants, such as studies on ancient humans, for which contamination from modern humans might still pose a challenge.

The observed difference in sequencing success, measured as final coverage, compared to previous studies can have several causes. First, even for lower sequencing depths, we still generated relatively high coverage (average coverage for predigestion and predigestion + bleach experiment: 41.42 bp/kb). Various studies (e.g., [19,23]) have reported findings based on much smaller efforts (e.g., average coverage in [23]: 0.3 bp/kb). However, such low amounts of endogenous sequencing reads may overestimate library complexity and thus expected final coverage, even when extrapolating complexity to higher sequencing efforts [40]. Second, sequencing success has been reported in different ways. Whereas some studies report the number of uniquely mapping fragments after duplicate removal [19,22], or endogenous content and/or complexity, we reported final coverage. These different definitions complicate efficiency comparisons. We consider final coverage to be a better proxy for sequencing success, since it directly captures the amount of data generated for a given sequencing effort, taking into account not only endogenous DNA content and complexity, but also DNA fragment length. Third, most of our samples were well preserved, resulting in relatively high endogenous contents regardless of treatment. It has already been reported that such high quality samples may show different patterns after decontamination pretreatments compared to low quality samples [21]. Finally, since ancient samples from other contexts may differ, for example in DNA fragmentation, damage rates, endogenous DNA content and higher presence of humic acids, the efficiency of different methodological approaches can vary between samples.

Bleach wash and predigestion affected DNA composition, damage patterns and endogenous DNA content in different ways. Predigestion resulted in a significant increase in endogenous DNA content, and a similar trend was observed for the bleach + predigestion treatment. Conversely, bleach wash followed by predigestion resulted in lower GC content and higher DNA damage, indicating that bleach wash affects DNA in a different way than predigestion. Concerns have been raised before that bleach wash may elevate damage patterns, and can even result in contaminating sequences developing DNA damage profiles identical to those found in ancient DNA [53], although this effect has not been found in other studies [20,22,51]. Importantly, despite elevated damage patterns due to bleach treatment, we found that overall damage patterns were still low in the samples after USER treatment. Remarkably, the two pretreatment methods showed a notable difference in endogenous content recovery success between samples, with some samples having over a two-fold increase in endogenous content, whereas other samples had similar or even lower endogenous content after pretreatment. These differences could potentially be explained by sample quality and/or contamination levels, although our sample sizes are too low to confirm this.

Surprisingly, the average fragment length was significantly higher for MinElute columns, although these columns are advertised as retaining shorter fragment lengths compared to QIAquick columns. Furthermore, GC content was significantly lower for samples extracted with MinElute columns, suggesting that MinElute columns show a bias toward retaining GC-poor DNA fragments. Nevertheless, the final coverage did not differ between the two treatments, potentially due to small stochastic differences in endogenous content and complexity. Larger sample sizes may arguably reveal a small but significant increase in coverage when using MinElute columns due to the larger average fragment lengths. These results remain speculative, however, and further systematic testing is necessary to test this hypothesis.

As hypothesized, the average fragment length retrieved with 10 kDa vivaspin filters was shorter than with 30 kDa vivaspin filters and resulted in a trend toward higher endogenous content. Furthermore, the probability of cytosine deamination in a single-stranded context was significantly higher in the 10 kDa vivaspin filter. Overall, these results suggest that centrifugal concentrator filters with lower MWCO values recover shorter, and consequently, more damaged DNA fragments. Nevertheless, we did not find a difference in final coverage between the two filters. One possible reason for this could be the fact that this experiment was tested with relatively well-preserved permafrost samples. A more pronounced difference in fragment length retrieval, and potentially also endogenous content, may be observed when working with more degraded samples. Finally, average retrieved fragment length of the final sequencing libraries is not only limited by the centrifugal concentrator filters, but also by clean-up steps with silica columns during extraction and library preparation.

We also compared whether halving the amount of USER enzyme had an effect on the DNA damage rate. Although the sample size was too small for statistical analyses, we did not observe any difference in coverage or DNA composition. All three samples had a slightly higher average fragment length when half the amount of USER was used. The probability of deamination damage in single-stranded context was also on average 11% higher with half the USER amount, although the overall probability was still below 10% for all samples in either of the two USER experiments. Reduced USER efficacy in terminal base pairs has been reported before [21,54,55], and we also find these patterns in our own samples for both control and reduced USER treatments. One workaround to avoid this problem, regardless of USER concentration, could be to trim the first and last base pair from the fragments, at the cost of reducing genome-wide coverage. Given the high cost of USER treatment (approximately one-third of the total costs to process a sample into a sequencing library when using 0.3 U/µL), halving the USER concentration seems warranted.

In summary, neither predigestion nor bleach wash followed by predigestion increased final sequencing coverage of our samples. We would consequently advise against routine use of these chemical pretreatments in extraction protocols, at least for ancient samples that are relatively well preserved (e.g., permafrost samples). Moreover, we tentatively conclude that using a lower USER concentration (0.15 U/µL) is still efficient enough to remove most DNA damage, and this would be an easy way to reduce the cost of DNA extractions. Our findings suggest no difference between MinElute and QIAquick columns for ancient DNA retrieval, although we hypothesize that the significantly higher average fragment length in MinElute columns may, on average, increase final coverage. Vivaspin columns with lower MWCO values retrieved shorter DNA fragments and showed a trend toward retrieving higher endogenous content. However, this trade-off resulted in no significant difference in sequencing efficiency. For well-preserved permafrost samples, the MWCO did not have a significant effect, although the lower MWCO might be more important when working with highly degraded samples that typically have shorter average fragment lengths. Based on these findings, we developed an optimized ancient DNA protocol that we expect will lead to improved DNA recovery from permafrost samples, while at the same time maintaining comparatively low costs and high time efficiency. This new protocol is presented in Appendix A.

## Figures and Tables

**Figure 1 genes-13-00687-f001:**
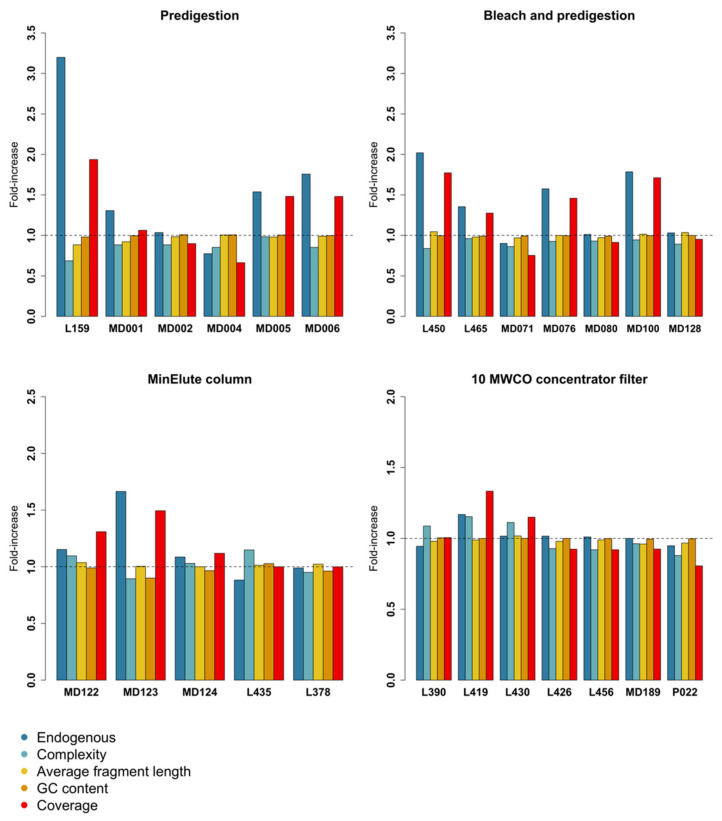
Relative efficiency of predigestion, bleach wash followed by predigestion, different silica columns, and centrifugal concentrator filters with different weight cut-offs, measured on each individual samples’ endogenous DNA content, complexity, average fragment length, GC content, and coverage. The dotted horizontal line marks the one-fold threshold. Bars below or above this threshold indicate that they performed worse or better, respectively, for that parameter compared to the control.

**Figure 2 genes-13-00687-f002:**
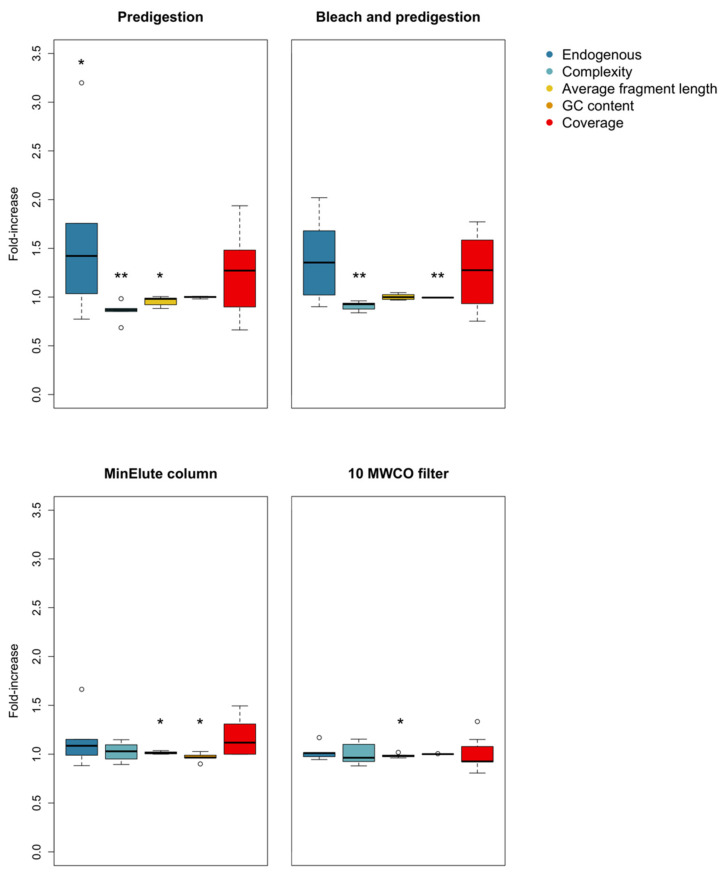
Boxplots of relative efficiency of predigestion, bleach wash followed by predigestion, different silica columns, and centrifugal concentrator filters with different weight cut-offs on (from left to right) endogenous DNA content, complexity, average fragment length, GC content, and coverage. Circles represent outlier values. Statistically significant results are marked with one (*p* < 0.05) or two (*p* < 0.01) asterisk(s). Whiskers represent 1.5 interquartile range, whereas boxes represent the first and third quartile. Values above one-fold indicate an increase in the treatment compared to the control for that parameter, whereas values under one-fold indicate a decrease.

**Figure 3 genes-13-00687-f003:**
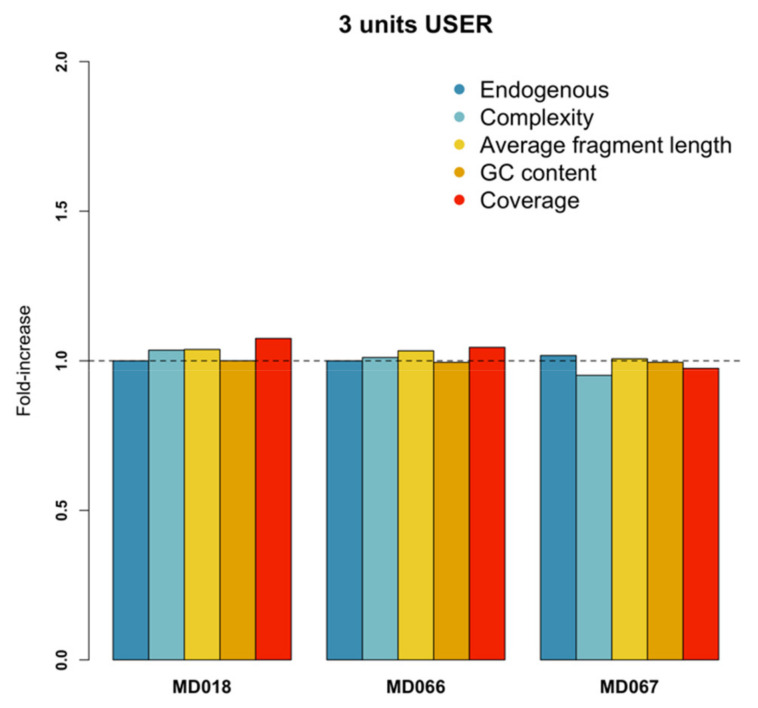
Relative efficiency of different USER concentrations on each samples’ endogenous DNA content, complexity, average fragment length, GC content, and coverage. The dotted horizontal line marks the one-fold threshold. Bars below or above this threshold indicate that the performance of the treatment was worse or better, respectively, for that parameter compared to the control.

**Figure 4 genes-13-00687-f004:**
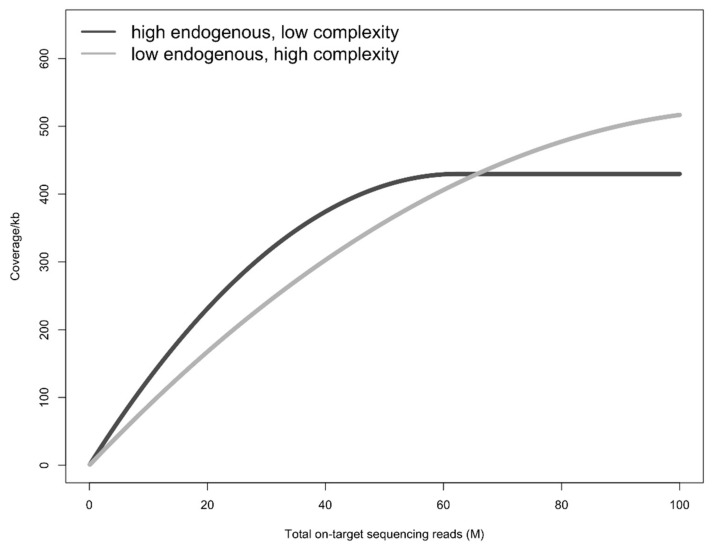
Conceptual figure to show how library complexity becomes more important at higher sequencing depths. The dark grey line depicts a library with high endogenous DNA content but low complexity, whereas the light grey depicts a library with low endogenous DNA content but high complexity. Although the high endogenous library results in higher coverage at lower sequencing depths (here <80 million on-target sequencing reads), the high complexity library becomes more efficient at higher sequencing depths.

**Table 1 genes-13-00687-t001:** Results of statistical tests.

		Paired Test	Test Statistic	*p* Value
Predigestion (*n* = 6)	Complexity	*t*-test	t = 4.061	0.00486
	Endogenous content	*t*-test	t = −3.135	0.01291
	Average fragment length	*t*-test	t = 2.019	0.04974
	GC content	Wilcoxon signed ranks	V = 9.5	0.6241
	Coverage	*t*-test	t = −1.562	0.08947
	Damage parameter δD	*t*-test	t = 1.633	0.08168
	Damage parameter δS	*t*-test	t = −0.009815	0.4963
	Damage parameter λ	*t*-test	t = 1.765	0.06888
Bleach wash and Predigestion (*n* = 7)	Complexity	*t*-test	t = 5.156	0.001052
	Endogenous content	*t*-test	t = −1.748	0.06555
	Average fragment length	*t*-test	t = −0.1617	0.4384
	GC content	*t*-test	t = 4.514	0.002023
	Coverage	*t*-test	t = −1.327	0.1164
	Damage parameter δD	*t*-test	t = −4.912	0.00134
	Damage parameter δD	*t*-test	t = −8.278	0.00008414
	Damage parameter λ	*t*-test	t = −3.797	0.004502
MinElute column (*n* = 5)	Endogenous content	Wilcoxon signed ranks	V = 7	0.5
	Complexity	*t*-test	t = −0.615	0.2859
	Average fragment length	*t*-test	t = −2.3524	0.03916
	GC content	Wilcoxon signed ranks	V = 15	0.03125
	Coverage	Wilcoxon signed ranks	V = 5	0.3125
	Damage parameter δD	Wilcoxon signed ranks	V = 7	0.5
	Damage parameter δS	Wilcoxon signed ranks	V = 12	0.1562
	Damage parameter λ	Wilcoxon signed ranks	V = 15	0.03125
10 MWCO filter (*n* = 7)	Endogenous content	*t*-test	t = −1.815	0.0597
	Complexity	*t*-test	t = −0.06492	0.47515
	Average fragment length	*t*-test	t = 2.002	0.0461
	GC content	Wilcoxon signed ranks	V = 1	0.5
	Coverage	Wilcoxon signed ranks	V = 18	0.28905
	Damage parameter δD	Wilcoxon signed ranks	V = 18	0.28905
	Damage parameter δS	Wilcoxon signed ranks	V = 25	0.03906
	Damage parameter λ	Wilcoxon signed ranks	V = 18	0.28905

## Data Availability

All individual read data are available at the European Nucleotide Archive (ENA, www.ebi.ac.uk/ena, accessed on 22 March 2022) under study Accession no. PRJEB51736.

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
