# Peer review of "Development and Optimization of a Silica Column-Based Extraction Protocol for Ancient DNA"

_genes, 2022, doi:10.3390/genes13040687_

Round 1
Reviewer 1 Report
This paper represents a comparison of a standard ancient DNA extraction protocol with five modifications to the protocol: a pre-digestion; bleach wash and predigestion; choice of silica columns; molecular weight cut-off (MWCO) of centrifugal concentrator filters; and concentration of USER enzyme. Efficacy of the treatments was measured by the amount of endogenous DNA (reads mapping to the reference genome divided by the total number of reads before duplicate removal), the complexity (proportion of uniquely mapped reads after duplicate removal), fragment length, GC content and coverage. The paper is very well written with thorough experimental design and statistical analysis. It should be of great interest to the ancient DNA community.
When aggregated over all samples, no significant differences in coverage were found and any increase in endogenous DNA was usually balanced by a reduction in complexity (Figure 2 and Table 1). However, there were differences amongst the individual samples (Figure 1). For example, modification of predigestion for L159 resulted in a 2-fold increase in coverage and > 3-fold increase in endogenous DNA. I think the authors should comment on the between sample variance (inter-bone differences). In my experience, this is the biggest challenge for extracting DNA from bones. For example, while the choice of MWCO had very little influence on any of the samples treated, predigestion (with and without bleach) had a far greater influence on the differences amongst samples. What is it about predigestion that has such a variable effect on different bones while MWCO has so little effect?
How were surfaces, sample vessels (eg. tubes), pipettes, etc cleaned or prepared to minimise the potential for exogenous DNA contamination?
Section 2.3.1 (Predigestion) involves addition of EDTA, urea and proteinase K to the powdered bone, as for the standard protocol. It is unclear if this was performed as well as (but prior) to addition of these in the standard protocol, or instead of these in the standard protocol. Could the authors make this explicit?
Section 2.3.2 (Bleach wash and predigestion): similar issue. After bleach washing, exactly which (and how many) pre-digestions were employed?
What may help is a diagram (figure) showing numbers of samples being directed through each of the protocol variations.
In Section 2.5 (Data processing & Analysis), “Complexity” is defined, but not on first use. Could the authors please define on first use?
In Section 2.5 (Data processing & Analysis), the authors say that they “downsampled” the sequencing files to match the lowest number of sequences for each pair of subsamples. I think this would have the effect of artificially dampening any differences in coverage which is probably why no significant differences were found (Table 1). I think the authors should comment on this.
I found it difficult to interpret Figure S11 (Damage plots generated with mapDamage v2.0.9, showing the base misincorporations in the 25 terminal 493 bases) and I wonder if the authors could explain each of the panels in this figure?
Reviewer 2 Report
In their work, Dehasque et al. present an ancient DNA silica-based extraction protocol to be used in palaeogenomics research. The work is not pretentious in itself, but it definitively provides useful information to people from different research areas. Moreover, the MS is exhaustive, well written and organized; as such it is easy and quick to read. The methodological approach, statistics included, is correct and clear to understand. The graphical items are effective, even if their quality (i.e., resolution) could be improved.
I found just a few oversights that the Authors might want to fix . For instance:
- In the author list, number “8” next to “Sergey Vartanyan should be reported as apex
- P 1, L 43: “as woolly mammoths and Neanderthals” provide the scientific names too, please
- P 1, L 65 remove full stops after citation “[27].”
- When you first cite a given brand (e.g. “Life Technologies”), provide also the country and town it is located.
- Pay attention not to mix British and American English wordings. For instance, at P 13, L 434 you used “hypothesised” (British), but then at L 469 you use “hypothesize” (American). Likewise, at P.11, L 381 you used “optimised”, but then at L 358 you used “optimized”. You also use “toward” and “towards”. Make a choice and be consistent, please.
